# Floating Epoch Length Improves the Accuracy of Accelerometry-Based Estimation of Coincident Oxygen Consumption

**DOI:** 10.3390/s24010076

**Published:** 2023-12-22

**Authors:** Henri Vähä-Ypyä, Pauliina Husu, Tommi Vasankari, Harri Sievänen

**Affiliations:** 1The UKK Institute for Health Promotion Research, 33500 Tampere, Finland; henri.vaha-ypya@ukkinstituutti.fi (H.V.-Y.); pauliina.husu@ukkinstituutti.fi (P.H.); tommi.vasankari@ukkinstituutti.fi (T.V.); 2Faculty of Medicine and Health Technology, Tampere University, 33014 Tampere, Finland

**Keywords:** accelerometer, energy consumption, measurement, movement, physical activity, prediction

## Abstract

Estimation of oxygen consumption (VO_2_) from accelerometer data is typically based on prediction equations developed in laboratory settings using steadily paced and controlled test activities. These equations may not capture the temporary changes in VO_2_ occurring in sporadic real-life physical activity. In this study, we introduced a novel floating epoch for accelerometer data analysis and hypothesized that an adaptive epoch length provides a more consistent estimation of VO_2_ in irregular activity conditions than a 6 s constant epoch. Two different activity tests were conducted: a progressive constant-speed test (CS) performed on a track and a 6 min back-and-forth walk test including accelerations and decelerations (AC/DC) performed as fast as possible. Twenty-nine adults performed the CS test, and sixty-one performed the AC/DC test. The data were collected using hip-worn accelerometers and a portable metabolic gas analyzer. General linear models were employed to create the prediction models for VO_2_ that were cross-validated using both data sets and epoch types as training and validation sets. The prediction equations based on the CS test or AC/DC test and 6 s epoch had excellent performance (R^2^ = 89%) for the CS test but poor performance for the AC/DC test (31%). Only the VO_2_ prediction equation based on the AC/DC test and the floating epoch had good performance (78%) for both tests. The overall accuracy of VO_2_ prediction is compromised with the constant length epoch, whereas the prediction model based on irregular acceleration data analyzed with a floating epoch provided consistent performance for both activities.

## 1. Introduction

Physical activity (PA) offers significant health benefits and mitigates health risks [1]. It is important to regularly measure population-level PA with valid methods. Such methods should be able to measure the frequency, duration, and intensity of PA, and desirably also the type of activity and its context [2]. These methods can be divided into self-reports and device-based methods. Self-reports provide pertinent information on the specific type or context of PA but are known to overestimate the exercise time while underestimating the active time accumulated during daily routines [3]. The device-based methods can assess PA in a more standardized manner regardless of the individual’s current fitness level and body weight, factors that may influence the subjectively perceived and reported intensity of PA [4]. In population studies, accelerometry is a commonly used cost-efficient and feasible method to collect dense data over long periods, allowing a detailed examination of daily behavior [2]. Also, accelerometry shows good compliance among participants [2] and provides relevant information on the dose–response relationship between physical behavior and health outcomes [1].

Accelerometer measurements of physical activity and estimations of its intensity are typically based on equations developed in laboratory settings using steadily paced activities [5]. However, they may offer little insight into how methods will perform in real-life conditions where activity and postural transitions are frequent and occur at irregular intervals [6]. Habitual PA comprises accelerations, turning, and braking, which all increase oxygen consumption (VO_2_) compared to constant-speed locomotion [7]. The extra VO_2_ depends on the rate of acceleration and deceleration and the number of changes in the direction of movement [8,9]. Changes in the movement speed or direction require applying additional force and energy to change the original momentum and kinetic energy to a new level [10].

The epoch length is the duration of time for which PA data is collected and processed for estimating coincident VO_2_. During continuous PA performed constantly at the same intensity, the accuracy of VO_2_ estimates from the accelerometer data is independent of the selected epoch length [11]. During intermitted conditions, short epoch lengths (less than 10 s) are effective in identifying PA and stationary phases, whereas longer epoch lengths (over 30 s) tend to smooth the acceleration signal and lead to missed short bursts of PA [5,11]. However, the longer epochs may be more appropriate for evaluating the physiological impacts of PA on health outcomes [11]. In a real-life context, different epoch lengths yield substantially different estimates of the total accumulated PA time, which can be confusing when the individual volumes of daily PA are evaluated [12].

Although short epoch lengths can detect short bursts of PA, they are not able to detect the higher energy expenditure caused by changes in the direction or speed of movement. A higher speed increases both the VO_2_ and accelerometer output but changing the body direction by 180°, i.e., performing a U-turn increases the VO_2_ while the accelerometer output does not respond similarly [7,13]. Current accelerometric prediction models cannot estimate the increased metabolic requirements caused by temporary or sporadic accelerations, decelerations, and rotations of the body, exceeding the metabolic requirement of constant-speed movements [9,14].

The accelerometer measurements of human movements can be contaminated with noise, which can be mitigated by applying a low-pass filter to the raw data [15]. The choice of the filter and its parameters depends on the noise characteristics and the desired trade-off between noise reduction and preserving meaningful motion data. For example, Wundersitz et al. (2015) found a low-pass filter with a cutoff of 12 Hz appropriate for measuring peak impacts during different types of movements [16]. Correspondingly, Fridolfsson et al. (2018) found that signal frequencies above 10 Hz did not include relevant information relative to movement [17]. Under normal conditions, the peak frequency rarely exceeds 6 Hz [18], and most relevant information ranges up to 4 Hz, corresponding to the step frequency [17].

In this paper, we introduce a novel approach to using a floating epoch length for estimating VO_2_ from raw triaxial accelerometer data and compare the performance of developed VO_2_ estimation models in walking tests, which were performed either at a constant speed or contained both acceleration and deceleration phases. The floating epoch algorithm estimates the change in the kinetic energy by setting the epoch length according to the coincident stride cycle. This approach can help ensure that all relevant parts of acceleration data are considered for analysis while providing the possibility to estimate the energetic cost of the changes in the kinetic energy. We hypothesize that the continuous adjustment of the epoch length improves the prediction of VO_2_ compared to the conventional fixed epoch length approach. Our objective is to develop a more accurate and comprehensive prediction equation for VO_2_ compared to existing prediction models.

## 2. Materials and Methods

### 2.1. Accelerometry

This study employed the raw acceleration signals from a triaxial accelerometer (Hookie AM20, Traxmeet Ltd., Espoo, Finland) collected during a constant-speed (CS) walking test and a walking test including both acceleration and deceleration phases (AD/DC). This accelerometer employs the commonly used 13-bit digital triaxial acceleration sensor module (ADXL345; Analog Devices Inc, Norwood, MA, USA). The measurement range of each measurement axis of the accelerometer is ±16 g (g denotes the Earth’s gravity, 9.81 m/s^2^), and the data is measured at a 100 Hz sampling frequency. The peak acceleration with hip-worn accelerometer can exceed 10 g in foot ground contacts during running [19] and a sampling frequency lower than 100 Hz can lead to missed movement events [20]. The accelerometer was attached to a hip-mounted elastic belt at the level of the iliac crest (Figure 1).

### 2.2. Test Protocols

The data employed in the present analysis were collected in our previous studies, where the participants were selected with convenience sampling [21,22]. The CS test was carried out at the premises of Vierumäki Sports Institute near the city of Lahti, Finland, and the participants were recruited from the Lahti region by the staff of the Vierumäki Sports Institute. The AC/DC test was carried out at the premises of the UKK Institute in the city of Tampere, Finland, and the participants were recruited from the Tampere region by the staff of the UKK Institute. Participants were informed about the experimental test protocol, and they gave their written informed consent before performing their test. This study conformed to the code of Ethics of the World Medical Association Declaration of Helsinki and was approved by the Regional Ethics Committee of the Expert Responsibility area of Tampere University Hospital with codes R11115, R12246, and R13040.

Before testing, participants’ body height and weight were measured with standard methods. During both the CS and AC/DC tests, VO_2_ was continuously measured with a portable respiratory gas analyzer (Oxycon Mobile, CareFusion, Yorba Linda, CA, USA), which employs a telemetry system for data collection. A metabolic cart is designed to evaluate the body’s energy expenditure and respiratory gas exchange. The device gathers and analyzes information about how the body utilizes oxygen and produces carbon dioxide during different activities. Before each test, the flow meter, oxygen sensor, and carbon dioxide sensor of the metabolic cart were calibrated against known volumes and contents of oxygen and carbon dioxide gases.

In the CS test, 29 adults (15 males and 14 females) performed a pace-conducted non-stop test on a 200 m track. The initial speed was 0.6 m/s, and the speed was increased by 0.4 m/s every 2.5 min. Each stage lasted 2.5 min. The pace was controlled by light sources alongside the track, which were automatically lit up one at a time according to the required pace (Figure 2a). The participant had to keep up with the light sources. All light sources were also visible to the test supervisor. The selected gait type was recorded for each stage, and only completely walked stages were included in the present analysis. As a measure of steady-stage VO_2_ for a given speed, the mean VO_2_ of the final minute of the corresponding stage was used. The acceleration was analyzed for the final 2 min of each stage [21].

In the AC/DC test, 61 adults (35 males and 27 females) walked back and forth on a 15 m track as fast as possible for 6 min (Figure 2b). Like the CS test, the steady-stage VO_2_ for a given speed was based on the mean VO_2_ of the final minute of the test. The acceleration was collected and analyzed throughout the test. Before the test, the participants familiarized themselves with the metabolic cart and the 15 m test track. The participants were instructed to walk back and forth around the cones as fast as possible for six minutes. At the end of every minute, the tester informed the passing of walking time and counted the number of laps each participant walked [22].

### 2.3. Data Analysis

The acceleration data was analyzed with constant 6 s epochs and floating epochs using the MAD algorithm [21]. The acceleration data was first preprocessed with a low-pass filter to remove the noise. Secondly, the data was bandpass filtered to extract the cycle length of movement, which sets the epoch length.

The length of the floating epoch was set by the stride time of the gait. A single stride time starts from the initial contact of one foot and continues to the subsequent contact of the same foot [23]. Thus, a single stride contains two steps, one with each leg. After analyzing the data within one epoch, the analysis window was then shifted forward by the duration of one step to begin analyzing the next epoch. The data analysis with the floating epoch was based on the MAD value of each epoch, the difference in the MAD value between the adjacent epochs, and the inverse of the step time (i.e., step frequency).

Tri-axial acceleration was measured from all three orthogonal measurement axes (*x*, *y*, and *z*) in actual g-units and stored for further analysis. In short, each measurement point (*i*) consisted of samples *x, y* and *z_._* The resultant acceleration (*r*), which defines the magnitude of the acceleration vector and contains both dynamic and static component of acceleration, was calculated for each (*i*) time point with Equation (1).
(1)$$r[i]=\sqrt{{x[i]}^{2}+{y[i]}^{2}+{z[i]}^{2}}$$

The resultant acceleration data was smoothed with a 2nd order low-pass filter with a 0.12 Hz cut-off frequency. The low-pass filtered value (*R_LF_*) for the time point (*i*) was calculated with Equation (2), where *R_LF_* is the low-pass filtered value of three adjacent points (*i, i −* 1 *and i −* 2), a is the filter coefficient (127/128 for the 0.12 Hz cutoff frequency) and *r*[*i*] is the resultant acceleration for the point (*i*).
(2)$$R_{LF}[i]=2aR_{LF}[i-1]-a^{2}R_{LF}[i-2]+(1-2a+a^{2})r[i]$$

The duration of the single step, defined by the start time point (j) and the number of included samples (N), was determined by an algorithm comprising two digital filters for removing low- and high-frequency noise and for detecting the zero-crossing. First, the resultant acceleration r was filtered with 2nd order Butterworth low-pass filter with a 12 Hz cutoff frequency (*R_LF_*_12_). The *R_LF_*_12_ for the time point (*i*) was calculated with Equation (3), where the filter coefficient value *a*_1_ is 46/512, *a*_2_ 93/512, *b*_1_ 125/128, and *b*_2_ 44/128. The accelerometer data filtered this way is demonstrated to have an acceptable level of concurrent validity compared to a 3-dimensional motion analysis system [16].
(3)$$R_{LF12}\left[i\right]=a_{1}r\left[i\right]+a_{2}r\left[i-1\right]+a_{1}r\left[i-2\right]+b_{1}R_{LF12}\left[i-1\right]-b_{2}R_{LF12}\left[i-2\right]$$

Secondly, the low-pass filtered data, *R_LF_*_12_, was bandpass filtered (*R_BP_*) with a 2nd order Butterworth filter with bandwidth from 1.0 Hz to 3.0 Hz for the time point (*i*) with Equation (4), where *a*_1_ is 16,139/2^22^, *a*_2_ 32,278/2^22^, *b*_1_ 15,565/4096, *b*_2_ 22,279/4096, *b*_3_ 14,239/4096, and *b_4_* 3429/4096.
(4)$$R_{BP}\left[i\right]=a_{1}R_{LF12}\left[i\right]-a_{2}R_{LF12}\left[i-2\right]+a_{1}R_{LF12}\left[i-4\right]+b_{1}R_{BP}\left[i-1\right]-b_{2}R_{BP}\left[i-2\right]+b_{3}R_{BP}\left[i-3\right]-b_{4}R_{BP}\left[i-4\right]$$

Finally, the duration of a single step time was determined by the zero-crossing method with hysteresis. The hysteresis was used to prevent artefacts or noise caused by false crossings. It involved two threshold levels: one for the rising edge (positive crossing threshold, 0.05 g) and another for the falling edge (negative crossing threshold, −0.05 g). Counting of the single step time started when the bandpass filtered acceleration value, R_BP_, went from negative to positive and crossed the rising edge threshold, 0.05 g level. The step time calculation could not stop until the signal value went below the falling edge threshold (−0.05 g) before the next positive zero crossing (0.05 g), i.e., step time, can be detected.

The MAD value of the *k*th step was calculated with Equation (5), where *N* is the number of samples in the step and *j* is the time point of the start of the step.
(5)$${MAD}_{k}=\frac{1}{N}\sum_{i=j}^{j+N-1}\left|r[i]-R_{LF}[i]\right|$$

The floating *MAD* value was calculated for two adjacent steps with Equation (6), where *MAD_f,k_* is the floating epoch *MAD*-value of the *k*th step, *MAD_k_* and *MAD_k_*_−1_ are the MAD values for the *k*th and preceding step, and *N_k_* and *N_k_*_−1_ is the number of samples in the *k*th and preceding step. Thus, for bipedal gait, each epoch contained successive left and right footsteps or vice versa. As the floating epoch had one step overlap, each step was analyzed twice.
(6)$${MAD}_{f,k}=\frac{{MAD}_{k-1}N_{k-1}+{MAD}_{k}N_{k}}{N_{k-1}+N_{k}}$$

The change in the *MAD* values between adjacent epochs (*dMAD*) was calculated with Equation (7), where *MAD_f,k_* and *MAD_f,k_*_−1_ are the *MAD* values of the two adjacent floating epochs windows.
(7)$${dMAD}_{f,k}=\left|{MAD}_{f,k}-{MAD}_{f,k-1}\right|$$

The step frequency, *f_s_*, for the *k*th epoch was calculated with Equation (8), where 100 Hz is the sampling frequency of the accelerometer.
(8)$$f_{s,k}=2\frac{100\;Hz}{N_{k-1}+N_{k}}$$

Figure 3a illustrates the calculation of the *MAD* values for single steps during a 6 s period of the CS test. The calculation of the stepwise *MAD* values is based on the use of Equations (1)–(5). Figure 3b illustrates the floating epoch *MAD* values, 6 s epoch *MAD* values, *dMAD* values and stride frequency during a 60 s period of the AC/DC test. The floating epoch *MAD* values can capture the turns during the AC/DC test in contrast to the 6 s epoch. The values were calculated using Equations (6)–(8).

### 2.4. Statistical Analysis

The VO_2_ prediction equations based on the constant 6 s epochs were determined from the data of the CS test and AC/DC test and validated with the AC/DC test and CS test, respectively. Similarly, the VO_2_ prediction equations based on the floating epochs were determined using both data sets as training and validation sets. Mean *MAD* values for the 6 s and floating epochs, *dMAD*, and *f_s_* were calculated for all stages of the CS test and for the entire AC/DC test.

Generalized linear models were employed to create prediction models for estimating VO_2_. Incident VO_2_ was the dependent variable, while the coincident *MAD* values based on the 6 s epochs or coincident *MAD* values, *dMAD* value, and *f_s_* based on the floating epoch served as the independent predictor variables. Coefficient of determination (R^2^), standard error of estimate (SEE), and mean absolute percent error (MAPE) were used to assess the predictive error of the prediction models. All statistical analyses were conducted using the statistics software (IBM SPSS Statistics for Windows, Version 29.0, Armonk, NY, USA)

## 3. Results

The participants’ characteristics are presented in Table 1. There were more participants in the AC/DC test than in the CS test. The proportion of males was somewhat higher in the AC/DC test while the number of males and females was almost equal in the CS test. The participants of the AC/DC test were on average 15 years older and 12 kg heavier than those of the CS test.

The VO_2_ values ranged from 8.7 to 30.2 mL/kg/min in the CS test and from 12.4 to 41.1 mL/kg/min in the AC/DC test. The mean values of the 6 s epoch *MAD* ranged from 51 to 660 mg, the floating *MAD* from 56 to 660 mg, *dMAD* from 5 to 27 mg, and step frequency from 0.54 to 1.20 Hz for the CS test. For the AC/DC test, the mean values of the 6 s epoch *MAD* ranged from 280 to 662 mg, the floating *MAD* from 281 to 662 mg, *dMAD* from 20 to 63 mg, and step frequency from 0.89 to 1.37 Hz. The Pearson correlation coefficient between the 6 s epoch *MAD* and the floating *MAD* was 0.9998 for both the CS and AC/DC tests.

Figure 4 illustrates the relationships between the measured VO_2_ and calculated parameters. The *MAD* values of the AC/DC test (6 s and floating epoch) tend to be smaller than the values of the CS test for VO_2_ values greater than 20 mL/kg/min. The *dMAD* values of the AC/DC test were higher than the values of the CS test.

Table 2 presents the VO_2_ prediction equations derived from the 6 s epoch data. Notably, the equation developed using the CS test demonstrated high predictive accuracy when applied to the CS test data but performed poorly when applied to the AC/DC test data. Conversely, the equation generated from the AC/DC test exhibited high predictive accuracy for the CS test data but performed poorly when applied to the AC/DC test data.

Table 3 presents the VO_2_ prediction equations for the floating epoch. The prediction was based on the *MAD* values, *dMAD* values and the exponent of the *f_s_*. Notably, the equation developed using the CS test demonstrated excellent predictive accuracy for the CS test but performed poorly when applied to the AC/DC test. In contrast, the equation derived from the AC/DC test demonstrated high predictive accuracy for both the AC/DC and CS tests.

Figure 5 illustrates the performance of the VO_2_ prediction models trained with the 6 s epochs and the CS test. The model underestimates over 25 mL/kg/min VO_2_ values for the AC/DC test. The Bland–Altman difference plot further shows that the limit of agreement is wider for the AC/DC test, indicating a greater variability in the predicted values. In addition, the Bland–Altman plot depicts an apparent trend in the AC/DC test results, as the difference between measured and predicted VO_2_ values tends to increase with increasing mean VO_2_ values.

Figure 6 illustrates the performance of the VO_2_ prediction model trained with the 6 s epoch and the AC/DC test. Although the results of the AC/DC test show no bias, there is an evident trend in the Bland–Altman plot. The difference between the measured and predicted VO_2_ tends to become larger as the mean VO_2_ increases. The CS test has a narrower limit of agreement in the Bland–Altman plot than the AC/DC test, but the model overestimates VO_2_ for the CS test.

Figure 7 illustrates the performance of the VO_2_ prediction equations trained with the floating epochs and the CS test. Notably, the model consistently underestimates VO_2_ values exceeding 25 mL/kg/min for the AC/DC test. The Bland–Altman difference plot further reveals that the AC/DC test exhibits a wider limit of agreement compared to the CS test, signifying a greater variability in predictions. Additionally, this plot demonstrates a clear trend in the AC/DC test results, with the difference between measured and predicted VO_2_ values increasing as the mean VO_2_ increases.

Figure 8 illustrates the performance of the VO_2_ prediction model trained using the floating epochs and the AC/DC test. Notably, the VO_2_ values are consistently underestimated for the CS test. The limits of agreement are similar for both the AC/DC test and CS test, suggesting a comparable predictive performance. Unlike the other models, the Bland–Altman difference plot for the AC/DC test displays no visible trend, indicating more stable results over a broad range of VO_2_.

## 4. Discussion

The VO_2_ prediction equation using the floating epoch and developed with the unsteady activity performed adequately with steady activity but not vice versa. The present findings support the notion that VO_2_ estimation models based on steady activities may contribute to the poor accuracy of VO_2_ estimations during daily activities [7]. Habitual activities comprise inherently sporadic and irregular patterns compared to steadily paced patterns occurring in walking or running for exercise. Therefore, the assessment of free-living PA could be more reliable with methods developed using irregular activities. Both models trained with the CS test data performed well on the training data (CS test) but poorly on the validation data (AC/DC test). This means that the prediction models based on steady activities are compromised by generalization problems. This kind of model may perform well on the training data but fails to transfer its predictive ability to new, unseen data [24,25]. This lack of generalization is likely to lead to poor performance in real-world measurements [26].

In the present study, the CS and AC/DC tests had different ranges of the measured VO_2_ values. The participants of the CS test walked at different speeds, while the participants of the AC/DC test were asked to walk as fast as possible. Thus, the AC/DC test did not contain low VO_2_ values typical for light PA and most daily habitual activities [27]. An increased range of walking speeds during the AC/DC test would have likely resulted in a better performance for the floating epoch model. In its present form, the models trained by the AC/DC test yielded narrow limits of agreement for the CS test with both the 6 s and floating epochs. However, both models, the 6 s and the floating epoch, produced slightly biased estimates of VO_2_.

The floating epoch length model improved the VO_2_ estimation by detecting the variations between steps due to braking and acceleration. Previously, shorter epoch lengths have been found to be more accurate for capturing the actual PA levels during intermittent PA than longer epochs [11]. Like the short epoch length, the floating epoch model can capture short bursts of PA. In addition, when calculating the change in the epoch-wise accelerometer output values, it became possible to estimate the extra VO_2_ due to braking and propulsive forces. Noteworthy, the floating epoch length model also had good performance for the steady activity of the CS test, where the change in the accelerometer output between the adjacent epochs was small. The different influence of the floating epoch and fixed epoch lengths on the predicted VO_2_ from a standing position to walking is illustrated in Figure 9. The predicted VO_2_ reached the highest values during the acceleration phase of walking with the floating epoch model, but during the steady walking pace, both models showed a comparable prediction of VO_2_.

The fixed epoch length can result in the over-smoothing of acceleration data and lead to a loss of potentially important details and features from the data [28]. In this study, the fixed, 6 s epoch length missed the changes in the *MAD* levels during braking and accelerating, which yielded underestimated VO_2_ values from 25 mL/kg/min upwards for the AC/DC test (Figure 3, Figure 4 and Figure 5). The largest difference between the measured and estimated VO_2_ value for the AC/DC test with the 6 s epoch was 17 mL/kg/min, when the CS test was used as the training set, and 13 mL/kg/min, when the AC/DC test was used as the training set. Correspondingly, the largest difference with the floating epoch was 15 mL/kg/min, when the CS test was used as the training set, and 7 mL/kg/min, when the AC/DC test was used as the training set.

Bipedal gait involves continuous fluctuations in energetics, differing from wheeled vehicles, like bicycles [29]. The increases in kinetic energy due to acceleration require extra positive mechanical work and the same applies to deceleration, where negative mechanical work is required to reduce the kinetic energy. The absence of energy fluctuations at the constant speed makes any speed changes energetically very expensive for wheeled vehicles, but on the contrary, for bipedal gait, limited alterations in the speed can be performed without a remarkable increase in metabolic energy consumption [29]. In this study, the *MAD* and VO_2_ values were at the same level for the AC/DC and CS tests up to the approximately 20 mL/kg/min level in terms of the VO_2_ (Figure 2). For higher intensities, the VO_2_ values were higher for the AC/DC test than for the CS test with the same *MAD* values. Therefore, the greatest benefits of the floating epoch model can be achieved with intensities over 20 mL/kg/min corresponding primarily to vigorously performed PA.

The limitation of this study is that the floating epoch model was tested only with walking and the speed range was narrow in the AC/DC test. The floating epoch model should also be tested with other than bipedal physical activities, like swimming, cycling, skiing, and gym training, and in transportation activities, like vehicle driving. The other activities will likely have an influence on the selection of appropriate filter characteristics. The filter type and cutoff frequency determine the level of smoothing applied to the raw accelerometer data and directly impact on the detection of the movement cycle patterns [30,31]. An excessively aggressive filter may result in over-smoothing, obscuring important high-frequency features and leading to missed or distorted cycle events, while an insufficiently smoothing filter can allow noise and fine-scale variations in the raw data to interfere with cycle detection, potentially leading to inaccuracies in movement detection and the classification of movements [32]. Therefore, the judicious choice of the filter is pivotal in achieving a precise and robust cycle detection, striking a balance between noise reduction and the preservation of essential information of coincident movements.

In free-living measurements, the floating epoch model can capture the pulsatile and sporadic activity behaviors. With fixed and longer, over 15 s epochs, the intensity of these kinds of movements becomes underestimated or can even be undetected and classified as stationary time [33,34]. On the other hand, a short interruption in PA can be picked up as a stationary behavior with shorter epoch lengths [34]. Therefore, the floating epoch will contribute to the accumulation of both stationary time and time spent in light, moderate, and vigorous PA. It is likely that the accumulated time of light PA is reallocated to other activities, if the accumulated time is calculated using the floating epochs instead of the fixed epoch length.

When employing floating epochs, the use of an analyzing window with a size that aligns with specific parameters of interest in coincident PA might be a reasonable approach. The time-course of physiological responses to acute PA depends on the various factors, including the type and duration of activity and individual fitness level [35]. The analyzing window size can modulate data smoothing, a long window size smooths data more than a short window size [36]. The longer analyzing window size is likely more suitable for responses with a longer time-course while it removes effectively short bursts of PA in the middle of stationary behavior or short breaks in the middle of a long steady PA period. Similarly, a short window size suits responses with a shorter time-course while it can capture shorts bursts of activity or breaks. A short window size can also improve the type of human activity recognition as the number of windows containing multiple activities can be minimized [37,38]. The optimal size of the analyzing window, i.e., the level of smoothing, depends on the goals of the PA measurement and analysis, and the trade-off between noise reduction and preserving meaningful information. These topics call for further research.

## 5. Conclusions

The floating epoch model can accurately capture the intensity of the short bursts of PA effectively and may thus enhance the performance of the laboratory-trained prediction models for VO_2_ in free-living conditions. This model shows a heightened accuracy in predicting spontaneous and sporadic PA, which is advantageous for research focused on child and adolescent PA or activities associated with occupational tasks. The assessment of PA in population studies may also benefit from this approach. Furthermore, its utility may extend to exercise interventions characterized by short, high-intensity intervals, establishing the floating epoch model as a valuable tool for advancing our understanding and application of PA-related interventions.

## Figures and Tables

**Figure 1 sensors-24-00076-f001:**
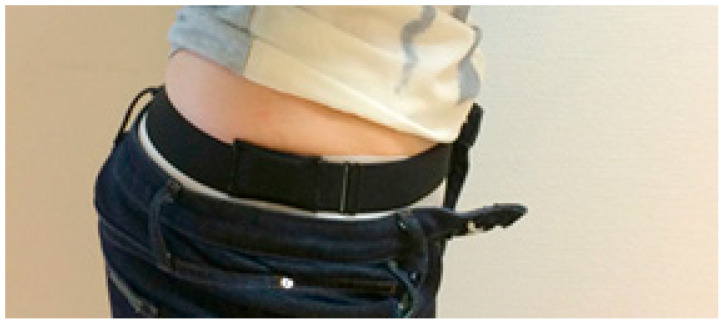
The placement of the accelerometer at the level of the iliac crest on an elastic hip-band.

**Figure 2 sensors-24-00076-f002:**
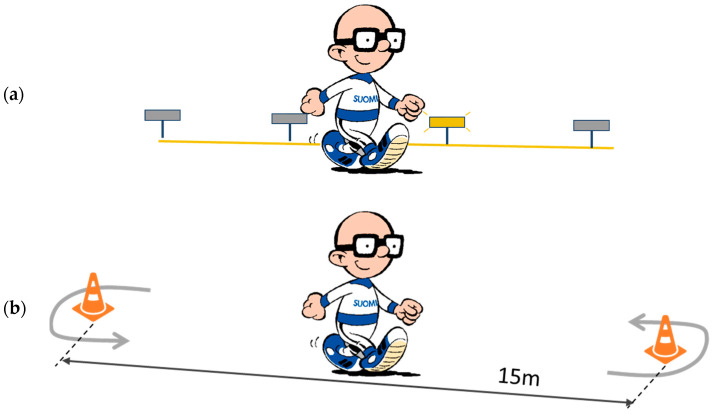
Illustration of the walking tests: (**a**) in the constant speed (CS) test, the participants were asked to follow the pace set by an advancing light source on a 200 m track; (**b**) in the acceleration/deceleration (AC/DC) test, the participants were asked to walk back and forth around the plastic cones on a 15 m track as fast as possible.

**Figure 3 sensors-24-00076-f003:**
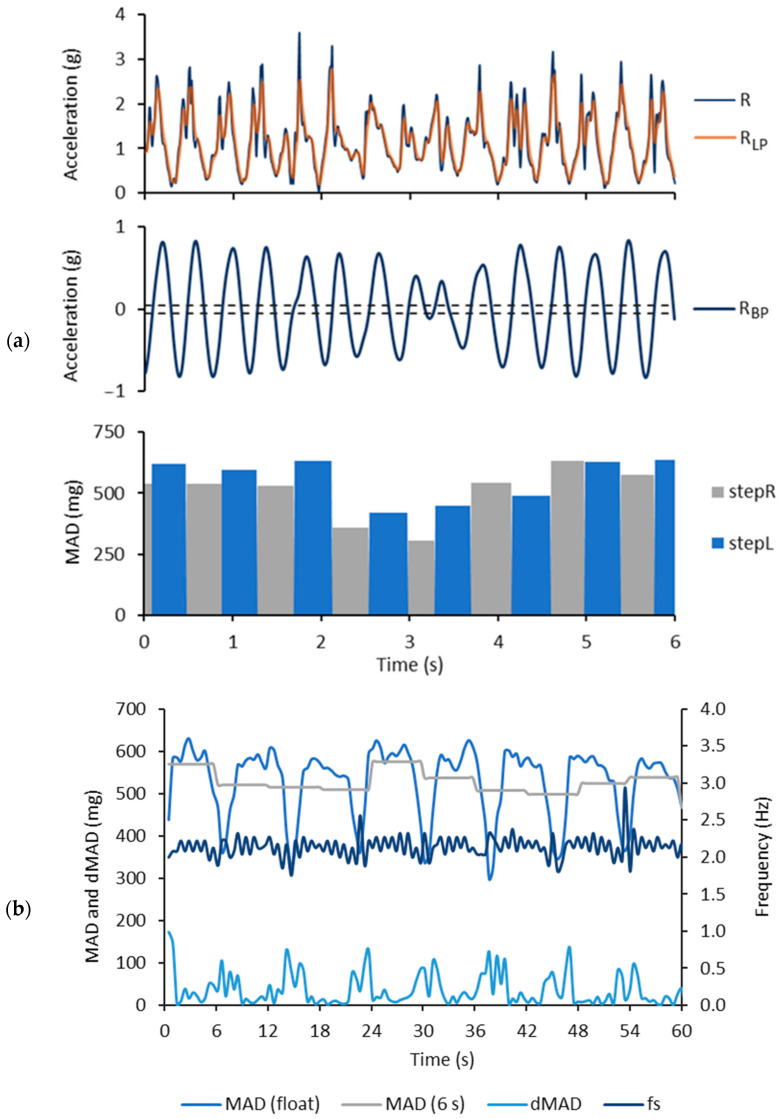
Example of measured and calculated parameter values during the AC/DC test: (**a**) The upper row shows the magnitude of the acceleration (*R*) and low-pass filtered acceleration (*R_LP_*) during the 6 s time containing one turn. The middle row shows the band-pass filtered acceleration (*R_BP_*). The dashed lines show the rising and falling edge thresholds for detecting the positive and negative crossings. The bottom row shows the mean amplitude deviation (*MAD*) value of each step. (**b**) The curves show the floating and 6 s epoch *MAD* values, change in the *MAD*-values between two adjacent floating epochs (*dMAD*) and step frequency (*f_s_*) during a 60 s time.

**Figure 4 sensors-24-00076-f004:**
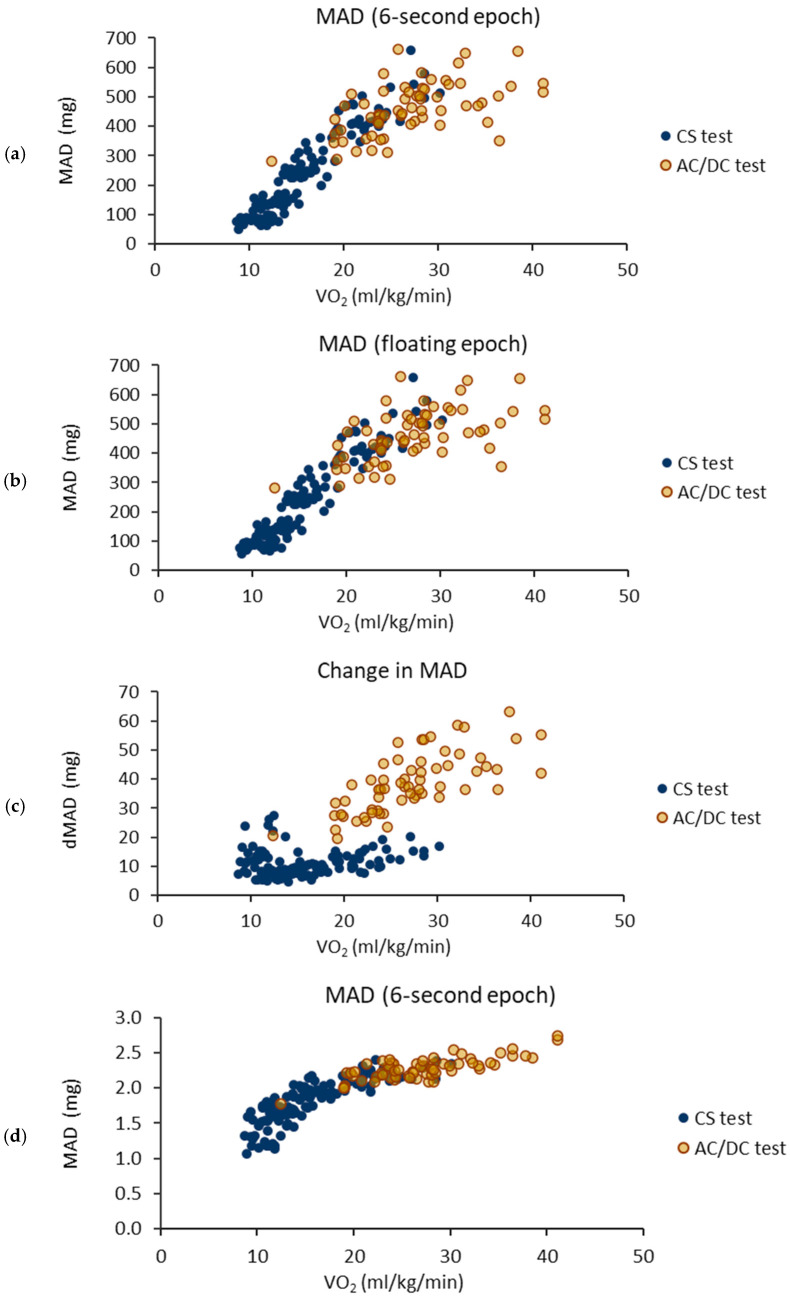
Mean values of measured accelerometer parameters for each participant as a function of coincident oxygen consumption (VO_2_) frequency during the acceleration and deceleration (AC/DC) test and constant speed (CS) test. (**a**) Mean amplitude deviation (*MAD*) for the 6 s epochs; (**b**) *MAD* for the floating epochs; (**c**) absolute change in *MAD* between the adjacent epochs (*dMAD*); and (**d**) step frequency.

**Figure 5 sensors-24-00076-f005:**
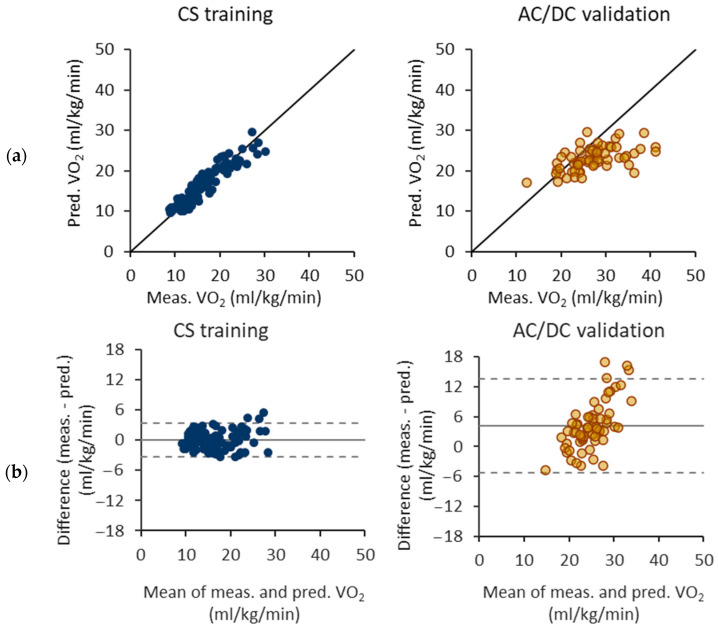
Performance of oxygen consumption (VO_2_) prediction equations trained with the 6 s epochs and constant speed (CS) test. (**a**) Measured and predicted VO_2_ for the CS test as a training set and the acceleration/deceleration (AC/DC) test as a validation set. (**b**) Respective Bland–Altman difference plots of measured and predicted VO_2_ values The solid line represents the mean difference and the dashed lines the 95% limits of agreement.

**Figure 6 sensors-24-00076-f006:**
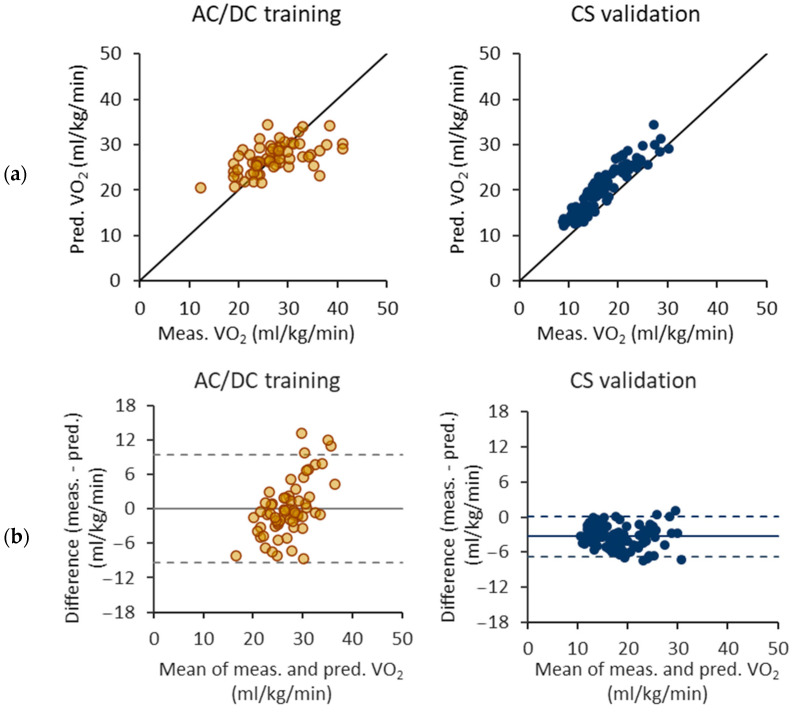
Performance of oxygen consumption (VO_2_) prediction equations trained with the 6 s epochs and acceleration/deceleration (AC/DC) test. (**a**) Measured and predicted VO_2_ for AC/DC test as a training set and constant speed (CS) test as a validation set. (**b**) Respective Bland–Altman difference plots of measured and predicted VO_2_ values. The solid line represents the mean difference and the dashed lines the 95% limits of agreement.

**Figure 7 sensors-24-00076-f007:**
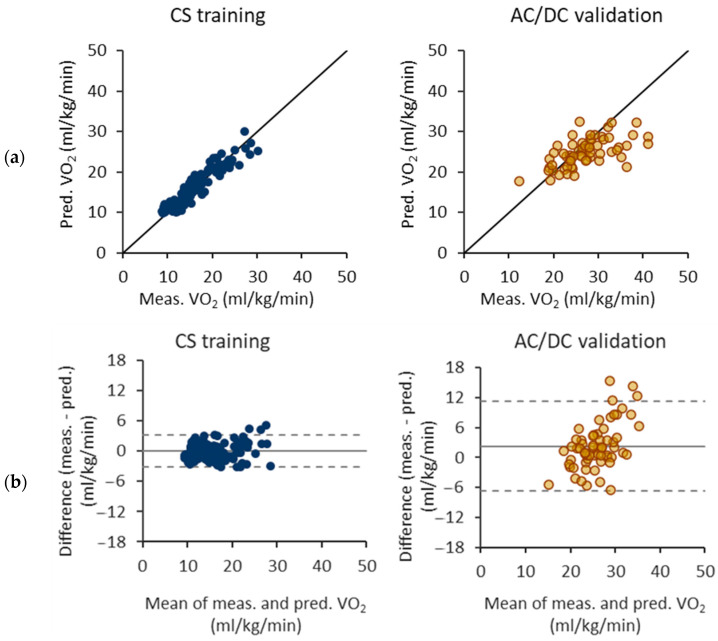
Performance of oxygen consumption (VO_2_) prediction equations with floating epochs and constant speed (CS) test. (**a**) Measured and predicted VO_2_ for CS test as a training set and AC/DC test as a validation set. (**b**) Respective Bland–Altman difference plot of measured and predicted VO_2_. The solid line represents the mean difference and the dashed lines the 95% limits of agreement.

**Figure 8 sensors-24-00076-f008:**
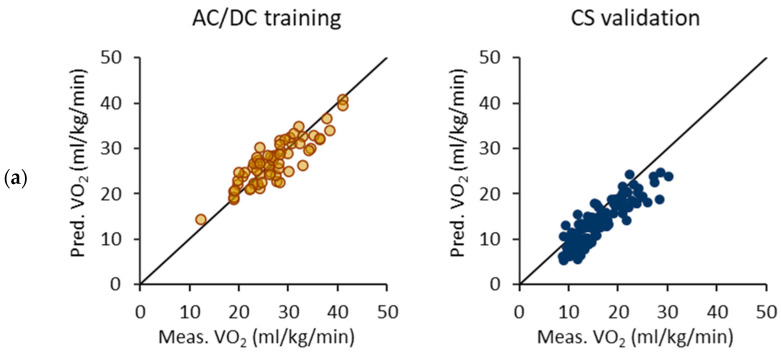
Performance of oxygen consumption (VO_2_) prediction equations trained with the floating epochs and acceleration/deceleration (AC/DC) test. (**a**) Measured and predicted VO_2_ for AC/DC test as a training set and constant speed (CS) test as a validation set. (**b**) Respective Bland–Altman difference plots of measured and predicted VO_2_ values. The solid line represents the mean difference and the dashed lines the 95% limits of agreement.

**Figure 9 sensors-24-00076-f009:**
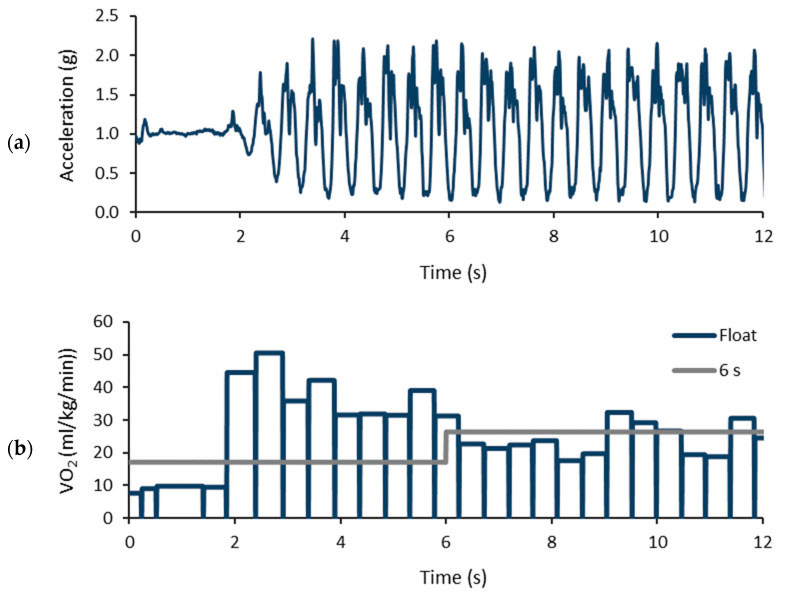
Illustration of a 12 s period containing a change in activity from standing to walking: (**a**) the magnitude of the measured acceleration; and (**b**) predicted oxygen consumptions using the floating epoch and 6 s epochs.

**Table 1 sensors-24-00076-t001:** Number of participants (N) and their background characteristics, mean (SD, range).

	AC/DC Test	CS Test
N	62	29
Male (%)	35 (56.5)	14 (48.3)
Female (%)	27 (43.5)	15 (51.7)
Age (years)	50.1 (12.5, 19.0–68.0)	36.6 (9.7, 20.0–59.0)
Height (cm)	172.9 (7.9, 154.0–191.0)	172.0 (10.0, 156.0–196.0)
Weight (kg)	82.6 (19.7, 52.4–146.8)	70.8 (12.7, 49.8–100.0)
BMI (kg/m^2^)	27.5 (5.4, 19.3–41.5)	23.5 (2.3, 19.3–29.6)

**Table 2 sensors-24-00076-t002:** Coefficients and performance of VO_2_ (mL/kg/min) prediction models based on 6 s epochs.

**Training Set**		**CS Test**	**AC/DC Test**
	Constant (*p*-value)	7.929 (<0.001)	10.379 (0.002)
	MAD (*p*-value)	0.033 (<0.001)	0.036 (<0.001)
	Bias	0	0
	SEE	1.68	4.81
	R^2^	88.8%	31.0%
	MAPE	8.7%	13.6%
**Validation Set**		**AC/DC Test**	**CS Test**
	Bias	4.13	−3.32
	SEE	4.82	1.76
	R^2^	31.0%	88.8%
	MAPE	16.9%	23.2%

CS test = constant-speed walking test, AC/DC test = acceleration and deceleration walking test, MAD = mean amplitude deviation, SEE = standard error of estimate, R^2^ = coefficient of determination, MAPE = mean absolute percentage error.

**Table 3 sensors-24-00076-t003:** Coefficients and performance of VO_2_ (ml/kg/min) prediction models based on the floating epoch.

**Training Set**		**CS Test**	**AC/DC Test**
	Constant (*p*-value)	7.186 (<0.001)	−3.160 (0.295)
	*MAD* (*p*-value)	0.033 (<0.001)	0.005 (0.585)
	*dMAD* (*p*-value)	0.068 (0.049)	0.218 (0.008)
	Exp(*f_s_*) (*p*-value)	−0.004 (0.986)	2.004 (<0.001)
	Bias	0.0	0.0
	SEE	1.64	2.9
	R^2^	89.3%	74.9%
	MAPE	8.4%	9.1%
**Validation Set**		**AC/DC Test**	**CS Test**
	Bias	2.29	2.56
	SEE	4.62	2.37
	R^2^	36.4%	78.2%
	MAPE	13.1%	19.6%

CS test = constant-speed walking test, AC/DC test = acceleration and deceleration walking test, *MAD* = mean amplitude deviation, *dMAD* = absolute change of MAD value between adjacent epochs, Exp = exponent, f_s_ = step frequency, SEE = standard error of estimate, R^2^ = coefficient of determination, MAPE = mean absolute percentage error.

## Data Availability

Non-identifiable data are available for research purposes from the corresponding author upon reasonable request.

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
