# Peer review of "Floating Epoch Length Improves the Accuracy of Accelerometry-Based Estimation of Coincident Oxygen Consumption"

_sensors, 2023, doi:10.3390/s24010076_

Round 1

Reviewer 1 Report

Comments and Suggestions for Authors

Thank you for the opportunity to review the manuscript titled ‘Floating epoch length improves the accuracy of accelerometry-based estimation of coincident oxygen consumption’. Well done to the authors for producing a very good piece of work.

I have one minor comment that needs to be addressed.

Results

In figure 1, top graph, RLP is difficult to view. Please can you find a way to highlight this curve to a greater degree.

Author Response

Reviewer 1

Comments and Suggestions for Authors

Thank you for the opportunity to review the manuscript titled ‘Floating epoch length improves the accuracy of accelerometry-based estimation of coincident oxygen consumption’. Well done to the authors for producing a very good piece of work.

I have one minor comment that needs to be addressed.

 Results

In figure 1, top graph, RLP is difficult to view. Please can you find a way to highlight this curve to a greater degree.

Thank you for your comment. We have changed the colors of the figure (now Figure 3 because of two new figures) and both lines should be now more clearly distinguishable.

Reviewer 2 Report

Comments and Suggestions for Authors

With regard to the manuscript: Floating epoch length improves the accuracy of accelerometry-2 based estimation of coincident oxygen consumption, submitted to Sensors.

The paper will contribute to knowledge and is worthy of publication. The general scope of the study appears to be acceptable and is of interest, but allow me to give you a few suggestions.

·  The authors do not explain the reason why they decided to indagate on accelerometer. Why not was studied other method? To demonstrate that accelerometer is a better alternative, an explanation of advantages and benefits must be exhibited in introduction. Please add some reference to support this.

·  The introduction provides an overview on estimation of oxygen consumption (VO2) from the accelerometer data. Habitual PA comprises accelerations, turning, and braking, which all increase oxygen consumption (VO2), however the currently used accelerometric prediction models cannot estimate the increased metabolic requirements caused by temporary or sporadic accelerations, decelerations, and rotations of the body.

·  The novelty regarding floating epoch length could be more explained and highlighted.

·  Line 85: The measurement range of the accelerometer is ±16g. Explain in more detail.  

·  Line 87: The accelerometer was attached to the hip-mounted elastic belt at the level of the iliac crest. Didactics would improve with the inclusion a figure/picture of a participant.

·  Line 92: VO2 was measured during these tests with a portable gas analyzer. Give more methodological informations.

·  Line 105: Does each stage last 2.5 minutes?

·  From 61 adults in AC/DC test, why only 29 were designated in CS test ? The lost is very huge. It would more interesting if the same participants had done the AC/DC and CS test.

·  Give more details of AC/DC test.

·  Didactics would improve with the inclusion a figure in discussion section (some scheme drawn by the authors) explaining their findings.

·  It is well written showing interesting interpretation. It is important to reader to know why floating epoch model (helping prediction of VO2) could be useful for science and for the medical/practical context. I would like this point of view to be more in-depth.

Author Response

Reviewer 2

Comments and Suggestions for Authors

With regard to the manuscript: Floating epoch length improves the accuracy of accelerometry-2 based estimation of coincident oxygen consumption, submitted to Sensors.

The paper will contribute to knowledge and is worthy of publication. The general scope of the study appears to be acceptable and is of interest, but allow me to give you a few suggestions.

 Thank you for your suggestions, which were useful to improve the manuscript. Below please find our responses to your comments.

  • The authors do not explain the reason why they decided to indagate on accelerometer. Why not was studied other method? To demonstrate that accelerometer is a better alternative, an explanation of advantages and benefits must be exhibited in introduction. Please add some reference to support this.

In large population studies focusing on physical activity and sedentary behavior, the accelerometer is feasible and commonly used and recommended method. Although it is not a perfect method, it provides useful measured information on individual physical behavior without subjective influence or recall bias. We wrote a new first paragraph to the Introduction on the above issues.

  • The introduction provides an overview on estimation of oxygen consumption (VO2) from the accelerometer data. Habitual PA comprises accelerations, turning, and braking, which all increase oxygen consumption (VO2), however the currently used accelerometric prediction models cannot estimate the increased metabolic requirements caused by temporary or sporadic accelerations, decelerations, and rotations of the body.
  • The novelty regarding floating epoch length could be more explained and highlighted.

We have added more text for the advantages and novelty of the floating epoch length in the last paragraph of Introduction. 

  • Line 85: The measurement range of the accelerometer is ±16g. Explain in more detail.

We have added further information and arguments on this issue in the Methods section 3.1 (1st paragraph). 

  • Line 87: The accelerometer was attached to the hip-mounted elastic belt at the level of the iliac crest. Didactics would improve with the inclusion a figure/picture of a participant.

We have added a new photo (Figure 1) showing the sensor placement is added to the Methods section (3.1).

  • Line 92: VO2 was measured during these tests with a portable gas analyzer. Give more methodological informations.

We have added more information about the metabolic cart to the Methods section 3.2 (2nd paragraph).

  • Line 105: Does each stage last 2.5 minutes?

Quite right. We have clarified this issue in the manuscript.

  • From 61 adults in AC/DC test, why only 29 were designated in CS test ? The lost is very huge. It would more interesting if the same participants had done the AC/DC and CS test.

We are sorry for this confusion.  In fact, there were no lost participants. The data for the present analysis were collected in previous studies. We have now clarified this issue in the Methods section 3.2. (1st paragraph). 

  • Give more details of AC/DC test.

We have now added more details about the AC/DC test in the Methods section 3.2 (4th paragraph) including a new figure (Figure 2).

  • Didactics would improve with the inclusion a figure in discussion section (some scheme drawn by the authors) explaining their findings.

Thank you for this request, which we found pertinent to clarify our findings. We added now a new figure (Figure 9) in the Discussion section, which shows the predicted VO2 based on a fixed 6-second epoch and the floating epoch from standing to walking. The predicted VO2 for the start of walking is higher with the floating epoch because of the need to increase the speed of the body compared to the 6-second epoch, which does not take this change in speed into account.

  • It is well written showing interesting interpretation. It is important to reader to know why floating epoch model (helping prediction of VO2) could be useful for science and for the medical/practical context. I would like this point of view to be more in-depth.

The conclusion part is now rewritten to emphasize this point of view more clearly.

Reviewer 3 Report

Comments and Suggestions for Authors

- Line 70-76: At the end of the introduction, despite the contextualization and the hypothesis formulated, the objective of the study was not explicit.

- Line 79: In this version presented in the Materials and Methods section, the authors did not describe aspects of the study (type, methodology, study location, how the population was recruited and selected).

- Particularly, a figure illustrating the execution of the tests, the flow and the sequence would bring more lucidity to the reading of this work. Not every analysis procedure is of global knowledge. As a suggestion, the insertion of a didactic illustration of the proposed protocol remains.

- Line 224: As the study design was not detailed, there was a lack of construction and presentation of a flowchart (Figure) to describe the process of selection, inclusion and participation in the study.

- Table 1 and Figure 2 are without a caption.

- What are the limitations of this study?

- What potential future contributions can be made from the results of this study?

Author Response

Reviewer 3

Comments and Suggestions for Authors

Thank you for the valuable comments. We have made improvements to the manuscript and below please find our responses to your comments.

- Line 70-76: At the end of the introduction, despite the contextualization and the hypothesis formulated, the objective of the study was not explicit.

The objective of the study is now described more clearly at the end of the Introduction.

- Line 79: In this version presented in the Materials and Methods section, the authors did not describe aspects of the study (type, methodology, study location, how the population was recruited and selected).

The participants were selected with convenience sampling close to the study locations. The study was conducted in two different locations. The details are now described in more detail in the Methods section 3.2. (1st paragraph).

- Particularly, a figure illustrating the execution of the tests, the flow and the sequence would bring more lucidity to the reading of this work. Not every analysis procedure is of global knowledge. As a suggestion, the insertion of a didactic illustration of the proposed protocol remains.

We have now added a new figure (Figure 2) illustrating the differences between the conducted tests.

- Line 224: As the study design was not detailed, there was a lack of construction and presentation of a flowchart (Figure) to describe the process of selection, inclusion and participation in the study.

As responded above, the study design is now better described in the Methods section.

- Table 1 and Figure 2 are without a caption.

The captions are added and completed.

- What are the limitations of this study?

The limitation of this study is now added to the Discussion (page 15, last paragraph)

- What potential future contributions can be made from the results of this study?

The conclusion paragraph is now rewritten to bring out potential contributions of using the floating epoch model.

Round 2

Reviewer 3 Report

Comments and Suggestions for Authors

Dear authors.

I congratulate you on the adjustments and improvements you have made to this work. It brought more rigor to the content presented.

But, Table 1 continues without the legend.

Apart from this small adjustment to make, I have nothing more to ask for addition.

Author Response

Thank you again for your constructive feedback on our work.

Table 1 was inadvertently split into two pages, the legend in one, and the remaining pages. Now Table 1 is on the same page.
